# Immunogenicity decay and case incidence six months post Sinovac-CoronaVac vaccine in autoimmune rheumatic diseases patients

The determination of durability and vaccine-associated protection is essential for booster doses strategies, however data on the stability of SARS-CoV-2 immunity are scarce. Here we assess anti-SARS-CoV-2 immunogenicity decay and incident cases six months after the 2$^{nd}$ dose of Sinovac-CoronaVac inactivated vaccine (D210) in 828 autoimmune rheumatic diseases patients compared with 207 age/sex-balanced control individuals. The primary outcome is the presence of anti-S1/S2 SARS-CoV-2 IgG at 6 months compared to 6 weeks after 2nd vaccine dose for decay evaluation. Secondary outcomes are presence of neutralizing antibodies, percent inhibition by neutralizing, geometric mean titers and cumulative incident cases at 6 months after 2nd dose. Anti-S1/S2 IgG positivity and titers reduce to 23.8% and 38% in patients ($p < 0.001$) during the six-month follow up and 20% and 51% in controls ($p < 0.001$), respectively. Neutralizing antibodies positivity and percent inhibition declines 41% and 54% in patients ($p < 0.001$) and 39.7% and 47% in controls (p < 0.001). Multivariate logistic regression analysis show males (OR = 0.56; 95% CI 0.40-0.79), prednisone (OR = 0.56; 95% CI 0.41-0.76), anti-TNF (OR = 0.66; 95% CI 0.45-0.96), abatacept (OR = 0.29; 95% CI 0.15-0.56) and rituximab (OR = 0.32; 95% CI 0.11-0.90) associate with a substantial reduction in IgG response at day 210 in patients. Although cellular immunity was not assessed, a decrease of COVID-19 cases (from 27.5 to 8.1/100 person-years; $p < 0.001$) is observed despite the concomitant emergence and spread of the Delta variant. Altogether we show a reduction in immunity 6-months of Sinovac-CoronaVac 2nd dose, particularly in males and those under immunosuppressives therapies, without a concomitant rise in COVID-19 cases. (CoronavRheum clinicaltrials.gov:NCT04754698).

Mass vaccination is the main measure to control the severe acute respiratory syndrome coronavirus 2 (SARS-CoV-2) spread and the emergence of new viral variants of concern[1]. While the pandemic drags on, the determination of immunogenicity durability is an essential step to establish booster dose strategies.

Data on the medium- and long-term persistence of immunity after vaccination against SARS-CoV-2 are scarce. Only a few cases series are reported with messenger RNA (mRNA) and viral vector vaccines, and the data have demonstrated a variable decline of antibody levels 2–6 months after two doses of SARS-CoV-2 vaccination in the general population[2–8]. In addition, a large prospective study in healthcare workers reported a substantial decrease of mRNA vaccine-induced antibodies by 6 months[8]. Some of these studies identified age and sex as associated with reduced durability of vaccine humoral immune response[3,8].

✉e-mail: eloisa.bonfa@hc.fm.usp.br

The inactivated Sinovac-CoronaVac vaccine is currently used in the most populated countries of the world, and its protective effect against hospitalization and death related to coronavirus infectious disease 2019 (COVID-19) was demonstrated in more than 10 million subjects[9]. The short-term waning of antibody response to this vaccine was evaluated in 159 healthcare workers with persistent seropositivity up to 98 days after vaccination, although with a significant reduction in antibody titers after 42 days[10].

The durability of vaccine immunity was also evaluated in a population of immunocompromised individuals composed of a few cancer patients under active therapy. The follow-up after full vaccination lasted solely 3–4 months. The study reported decay of IgG titers or inability to sustain IgG levels above the threshold[11,12]. With regard to autoimmune diseases, one study assessed 242 patients with a wide range of different conditions using a general computer-based questionnaire. They identified that participants with immunosuppression had a 65% reduction in IgG levels and 70% in neutralizing antibody (NAb) concentrations compared to those without these therapies, up to 6 months after vaccination with mRNA vaccine[8].

The deleterious impact of immunosuppressive therapy in a large autoimmune rheumatic diseases (ARD) population was reported for primary Sinovac-CoronaVac vaccination in a prospective study[13,14]. However, there is no report evaluating the long-term durability of anti-SARS-CoV-2 immunity in COVID-19 vaccinated ARD patients.

Here, we described the analysis of a large ARD population, that was conducted to assess prospectively the 6-month durability of SARS-CoV-2 immunity in fully vaccinated adults with Sinovac-CoronaVac compared with age- and sex-balanced control individuals without the rheumatic disease. We further evaluated incident symptomatic COVID-19 cases confirmed by real-time reverse transcriptase-polymerase chain reaction (RT-PCR). We also assessed risk factors for reduced 6-month durability of anti-SARS-CoV-2 immunity.

## Results

### Study design and participants

Participants originated from a large single center (Sao Paulo, Brazil) phase 4 controlled prospective study (no. NCT04754698, CoronavRheum) of immunogenicity and safety of two doses of Sinovac-CoronaVac vaccine in ARD patients and control group (CG)[13]. After applying the exclusion criteria, the final study groups consisted of 828 ARD patients and 207 healthy controls vaccinated with two doses (Fig. 1). ARD group included patients with: 27.5% (n = 228) rheumatoid arthritis (RA), 24.9% (n = 206) systemic lupus erythematosus (SLE), 23.7% (n = 196) axial spondyloarthritis, 6.2% (n = 51) primary vasculitis, 4.8% (n = 40) idiopathic inflammatory myopathies, 4.3% (n = 36) systemic sclerosis (SSc), and 4.3% (n = 36) primary Sjögren's syndrome. Regarding ARD current therapy at 6 weeks after the second dose (D69) the most frequently used were: 62.0% (n = 513) immunosuppressive drugs [209 (25.2%) methotrexate (MTX), 115 (13.9%) leflunomide (LEF), 109 (13.2%) mycophenolate mofetil (MFF), 90 (10.9%) azathioprine (AZA), 20 (2.4%) tofacitinib, 10 (1.2%) tacrolimus, 9 (1.1%) cyclosporine and 6 (0.7%) cyclophosphamide (CYC)], 36.5% (n = 302) prednisone (PRED) [median dose 5 (IQR 0.5–10) mg/day] and 35.7% (n = 296) biologic therapy [126 (15.2%) TNF inhibitor (35/126 combined to MTX), 46 (5.6%) tocilizumab (TCZ), 43 (5.2%) abatacept (ABA), 34 (4.1%) secukinumab, 26 (3.1%) belimumab, 17 (2.1%) rituximab (RTX), and 4 (0.5%) ustekinumab]. ARD and CG groups were comparable regarding median current age (p = 0.898), female sex (p > 0.999), and Caucasian ethnicity (p = 0.163) (Table 1).

### Anti-SARS-CoV-2 S1/S2 IgG responses in ARD patients and CG

Anti-SARS-CoV-2 S1/S2 IgG assessed at D0, D69 (6 weeks after vaccine second dose), and D210 (6 months after the 2nd dose) are presented in Table 2. At D0, ARD patients had significantly lower anti-S1/S2 IgG seropositivity compared to CG [142 (17.1%) vs. 68 (32.9%), p < 0.001].

From D69 to D210 (6 months after second dose), anti-S1/S2 IgG seropositivity rates reduced by 23.8% in ARD [650 (78.5%) vs. 495 (59.8%), p < 0.001] and 20% in CG [202 (97.6%) vs. 161 (77.8%), p < 0.001], with moderate but lower IgG persistence in ARD compared to CG at D210 (p < 0.001). IgG GMT from D69 to D210 declined significantly after the second dose in ARD [41.8 (38.0–46.0) vs. 26.1 (23.2–29.4) AU/mL, p < 0.001] and in CG [99.6 (88.2–112.6) vs. 48.8 (40.3–59.0) AU/mL, p < 0.001], with lower IgG levels in the former group at D210 (p < 0.001) (Table 2). The decrease in IgG titer, calculated as 1–Ln(IgG210/IgGD69), was significantly lower in ARD compared to CG [38% (95% CI 32–43%) vs. 51% (95% CI 43–58%), p = 0.004].

A subanalysis of ARD patients with positive anti-SARS-CoV-2 serology before vaccination demonstrated a significant increase in anti-S1/S2 GMT from D0 to D69 (p < 0.001) with a subsequent decrease from D69 to D210 (p < 0.001) [74.7 (95% CI 64.4–86.6) vs. 165.7 (95% CI 143.2–191.7) vs. 104.8 (95% CI 88.5–124.2) AU/mL, p < 0.001].

### Neutralizing antibodies responses in ARD patients and CG

From D69 to D210, NAb positivity declined by 41% in ARD [539 (65.1%) vs. 318 (38.4%), p < 0.001] and 39.7% in CG [181 (87.4%) vs. 109 (52.7%), p < 0.001]. There was a significantly greater reduction in NAb percent inhibition in those with positive NAb titers at D69 in the ARD vs. the CG [54% (95% CI 51–57%) vs. 47% (95% CI 41–52%), p = 0.024].

### Factors associated with negative IgG 6 months after second dose

At D210, negative anti-S1/S2 IgG in ARD group was associated with older age (p = 0.001), lower frequencies of SLE (p = 0.005) and SSc (p = 0.024) and higher frequency of male sex (p = 0.007) and RA (p < 0.001). Regarding the influence of current therapy, patients seronegative for IgG at D210 were more often under PRED (43.7% vs. 31.1%, p < 0.001), at higher median dose [7.5 (5–10) vs. 5 (5–10) mg/day, p = 0.043], anti-TNF (19.2% vs. 13.2%, p = 0.012), ABA (9.8% vs. 2.7%, p < 0.001), and RTX (3% vs. 1.1%, p = 0.028), while hydroxychloroquine (HCQ) use was more frequent in ARD with positive anti-S1/S2 IgG (Table 3). Multivariate logistic regression analysis using IgG positivity at D210 as the dependent variable and as independent variables those with p < 0.2 in univariate analysis (current age, male sex, RA, SLE, SSc, HCQ, sulfasalazine, PRED, LFN, anti-TNF, ABA, and RTX use), revealed that male sex (OR = 0.56, 95% CI 0.40–0.79, p < 0.001), PRED use (OR = 0.56; 95% CI 0.41–0.76, p < 0.001), anti-TNF use (OR = 0.66; 95% CI 0.45–0.96, p = 0.031), ABA use (OR = 0.29; 95% CI 0.15–0.56, p < 0.001), and RTX use (OR = 0.32; 95% CI 0.11–0.90, p = 0.031) were significantly associated with absence of anti-S1/S2 IgG 6 months after vaccination in ARD patients (Table 4).

### Factors associated with negative NAb 6 months after 2nd dose

For NAb analysis at D210, SLE diagnosis was less frequent in seronegative ARD patients (p = 0.019) whereas biologic therapy (p = 0.031), particularly ABA use (p = 0.018) was higher in patients without NAb (Table 3). After multivariate logistic regression analysis using NAb positivity as the dependent variable and as independent variables, those with p < 0.2 in univariate analysis (RA, SLE, sulfasalazine, anti-TNF, ABA, TCZ, and RTX use), only ABA use (OR = 0.48; 95% CI 0.24–0.97, p = 0.041) remained significantly associated with the absence of NAb at D210 in ARD patients (Table 4).

### Kaplan–Meier curve of COVID-19 cases and hospitalizations

Analysis of incident cases of RT-PCR confirmed COVID-19 was performed from the 1st dose to 10 days after the second dose (T1) and thereafter to 180 days after the second dose (T2). Evaluation of incident cases among ARD patients (n = 1193) revealed n = 79 cases (n = 33 in T1 and n = 46 in T2) of COVID-19 with n = 14 hospitalizations (n = 7 in T1 and n = 7 in T2) and n = 4 deaths (n = 1 in T1 and n = 3 in T2). The incident symptomatic COVID-19 cases reduced from 27.5 (95% CI 18.9–38.6)/100 person-years in T1 to 8.1 (95% CI 6.0–10.9)/100 person-

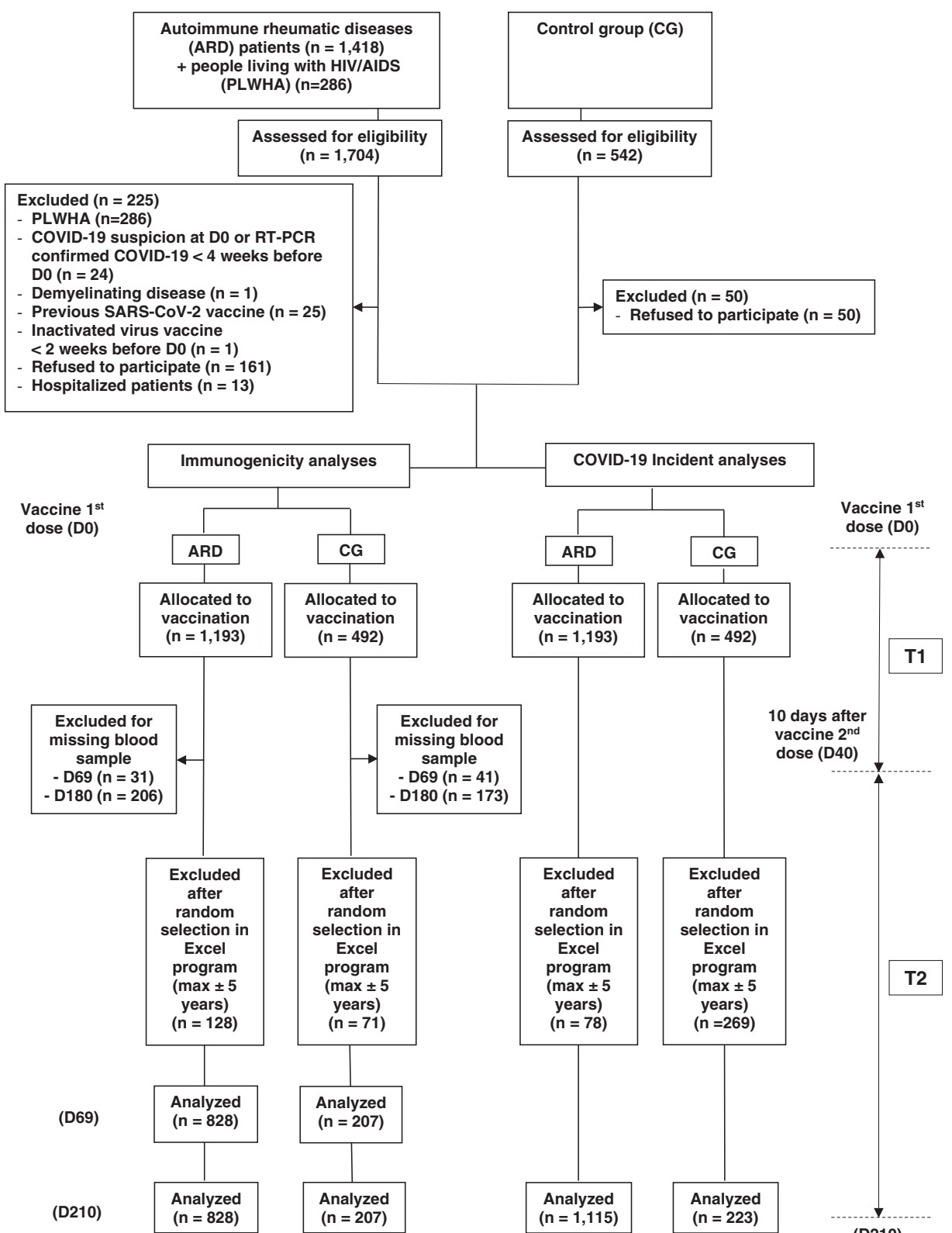

**Fig. 1 | Study flowchart.** The figure depicts enrollment, exclusions, and analysis of Anti-SARS-CoV-2 antibodies decay and COVID-19 incidence in patients with autoimmune rheumatic disease (ARD) and controls (CG). The participants allocated to vaccination were followed from the day of inclusion, which is the day of first dose vaccination with CoronaVac vaccine (DO) up to 10 days after the second dose vaccination (T1). The second period of follow-up (T2) ranged from T1 to 6 months after the second dose. PLWHA people living with HIV/AIDS, RT-PCR real-time polymerase chain reaction, SARS-CoV-2 Severe Acute Respiratory Syndrome Coronavirus 2, D0 inclusion of participants and vaccine first dose, D40 10 days after the vaccine second dose, D69 6 weeks after the vaccine second dose, D210 6 months after the vaccine second dose.

**Table 1 | Characteristics of patients with ARD and CG at 6 weeks after second Sinovac-CoronaVac (D69)**

| | ARD (n = 828) | CG (n = 207) | p-value |
|---|---|---|---|
| **Demographics** | | | |
| Current age, years | 51 (41–60) | 51 (42–60) | 0.898 |
| Age ≥60 years | 213 (25.7) | 54 (26.1) | 0.915 |
| Male sex | 204 (24.6) | 51 (24.6) | >0.999 |
| Caucasian ethnicity | 425 (51.3) | 95 (45.9) | 0.163 |
| **ARD** | | | |
| Rheumatoid arthritis | 228 (27.5) | — | — |
| Axial spondyloarthritis | 196 (23.7) | — | — |
| Systemic lupus erythematosus | 206 (24.9) | — | — |
| Primary vasculitis | 51 (6.2) | — | — |
| Systemic sclerosis | 36 (4.3) | — | — |
| Primary Sjögren syndrome | 36 (4.3) | — | — |
| Idiopathic inflammatory myopathy | 40 (4.8) | — | — |
| **Current therapy** | | | |
| Prednisone | 302 (36.5) | — | — |
| Prednisone dose, mg | 5 (5–40) | — | — |
| Prednisone ≥10 mg/day | 116 (14.0) | | |
| Prednisone ≥20 mg/day | 27 (3.3) | — | — |
| Hydroxychloroquine | 239 (28.9) | — | — |
| Sulfasalazine | 72 (8.7) | — | — |
| Immunosuppressive drugs | 513 (62.0) | — | — |
| Methotrexate | 209 (25.2) | — | — |
| Leflunomide | 115 (13.9 | — | — |
| Mycophenolate mofetil | 109 (13.2) | — | — |
| Azathioprine | 90 (10.9) | — | — |
| Tofacitinib | 20 (2.4) | — | — |
| Cyclophosphamide | 6 (0.7) | — | — |
| Tacrolimus | 10 (1.2) | — | — |
| Cyclosporine | 9 (1.1) | — | — |
| Biologic therapy | 296 (35.7) | — | — |
| TNF inhibitor | 126 (15.2) | — | — |
| Abatacept | 43 (5.2) | — | — |
| Tocilizumab | 46 (5.6) | — | — |
| Belimumab | 26 (3.1) | — | — |
| Secukinumab | 34 (4.1) | — | — |
| Rituximab | 17 (2.1) | — | — |
| Ustekinumab | 4 (0.5) | — | — |

Results are expressed in medians (interquartile range) and n (%). Continuous data were compared using Mann-–Whitney test, and categorical variables with the chi-square or Fisher's exact tests, as appropriate, always as two-sided analyses, without adjustments for multiple comparisons.

*ARD* autoimmune rheumatic diseases, *CG* control group.

years in T2, with an incidence rate decline of 19.4 (95% CI 12.7–26.1) cases/100 person-years ($p < 0.001$). Likewise, in the same time periods, hospitalizations due to COVID-19 decreased from 5.8 (95% CI 2.3–12.0)/ 100 person-years to 1.2 (95% CI 0.5–2.6)/100 person-years, with an incidence rate decline of 4.6 (95% CI 1.8–7.4) hospitalizations/100 person-years ($p = 0.001$).

Deaths due to COVID-19 occurred in $n = 4$ ARD patients: $n = 1$ in T1 (before the second vaccine dose) and $n = 3$ in T2 ($p = 0.694$), after the second dose of Sinovac-CoronaVac. Among these three later patients, two were negative for anti-S1/S2 IgG and NAb at D69, and only one was positive for both at moderate levels.

Further comparative survival analysis of ARD and CG ($p = 0.152$), with a random selection of age and sex comparable subjects, pointed

that $n = 33$ (ARD $n = 31$ and CG $n = 2$) RT-PCR confirmed COVID-19 incident symptomatic cases were reported during T1 (40 days) and $n = 52$ cases (ARD $n = 44$ in CG $n = 8$) during T2 (180 days) evenly distributed along this period in ARD patients (Fig. 2).

## Discussion

The present study demonstrated a substantial decline of anti-SARS-CoV-2 immunogenicity in ARD patients 6 months after the full inactivated vaccine schedule without a simultaneous increase in breakthrough cases. We further identified male sex and immunosuppression as deleterious for long-term antibody persistence in these patients.

A large number of ARD patients with a balanced age and sex CG included in this study was the main strength since it provided a unique opportunity to define more accurately risk factors for vaccine-induced anti-SARS-CoV-2 durability. In fact, age and sex were identified previously as determinants of the inability to sustain SARS-CoV-2 antibody levels in health population[3]. The use of internationally well-established disease classification criteria for the diagnosis of each disease in patients included in the study allowed a more accurate definition of the impact of different conditions and their therapy. The inclusion of a broad spectrum of non-rheumatologic immunosuppressed participants and the use of a generic computer-based questionnaire about existing conditions and treatment precluded a definitive conclusion about the subgroup of ARD and their treatment in a previous study[8].

The period of 6 months with a parallel observation of incident cases is endorsed by the report of breakthrough infection 4-month post-BTN162b2 vaccination associated with reduced levels of antibodies peri-infection and a significantly diminished humoral response in 6 months[1,8]. The uniform post-vaccination follow-up was an essential parameter for a more accurate definition of vaccine-induced antibody persistence at 6 months. In fact, previous studies have demonstrated that vaccine humoral response waning dynamics vary over time and are also distinct for IgG and NAb level[8]. An important limitation of the present study is the non-assessment of cellular immunity that also contributes to vaccine efficacy in this population. Even though NAb evaluated herein were reported to be associated with protective immunity, they do not surpass the T-cell evaluation[15].

We provide data of a reduction of anti-SARS-CoV-2 antibodies positivity over 6 months post second dose of Sinovac-CoronaVac in ARD patients with a magnitude of ~20% for patients and controls. We also confirmed herein with CoronaVac vaccination the previously reported observation of a robust (≥4 times) decrease in SARS-CoV-2 mRNA vaccine IgG levels after 6 months in healthcare workers[8]. We further demonstrated that the same phenomena, although less intense, occurred for the Sinovac-CoronoVac with a more expressive decrease in controls (51%) than in ARD patients (38%) at 6 months post second dose. In fact, a substantial decrease in anti-SARS-CoV-2 IgG titers has been previously reported in the healthy population[4,8]. In spite of that, the decrease in the IgG seropositivity rates was similar in both groups (24% in ARD and 20% in the control group) herein.

The analysis of NAb positivity, reported being a strong correlate of protection[15], revealed ~40% reduction for ARD patients and controls with a parallel waning of 54% in percent inhibition by NAb, after 6 months of Sinovac-CoronaVac vaccination. A more substantial decrease of 70% in NAb titer was reported 6-month post-mRNA vaccination for participants with immunosuppression[8]. This finding may be explained by the distinct immunocompromised populations analyzed in both studies. Regarding deleterious factors for vaccine-induced immunogenicity durability, we have identified in the multivariate logistic regression analysis that male sex, PRED and anti-TNF had a moderate impact on immunogenicity persistence, whereas a major harmful effect was evidenced for ABA and RTX after 6 months. These results are in line with previous evidence of our group demonstrating that PRED, MTX, anti-TNF, ABA, and RTX had the greatest negative impact on vaccine-induced anti-S1/S2 IgG response[3].

**Table 2 | Anti-SARS-CoV-2 S1/S2 IgG seropositivity (SP) rates and titers 6 weeks (D69) and 6 months (D210) after second dose of Sinovac-CoronaVac vaccination in patients with ARD in comparison to CG**

| | Seropositivity | | | Geometric mean titer (GMT) | | |
|---|---|---|---|---|---|---|
| | D69 | D210 | P-value (D69 vs. D210) | D69 | D210 | P-value (D69 vs. D210) |
| ARD, n = 828 | 650 (78.5) | 495 (59.8) | <0.001 | 41.8 (38.0–46.0) | 26.1 (23.2–29.4) | <0.001 |
| CG, n = 207 | 202 (97.6) | 161 (77.8) | <0.001 | 99.6 (88.2–112.6) | 48.8 (40.3–59.0) | <0.001 |
| P-value (ARD vs.CG) | <0.001 | <0.001 | | <0.001 | <0.001 | |

Seropositivity was defined as post-vaccination titer ≥15 AU/mL—Indirect ELISA (LIAISON® SARS-CoV-2 S1/S2 IgG, DiaSorin, Italy). GMT—Geometric mean titers (AU/mL); Frequencies of seropositivity are presented as number (%) and they were compared between groups (ARD and CG) and between timepoints (D69 vs. D210) using generalized estimating equations (GEE) with binomial distribution and logit link function, assuming autoregressive correlation matrix between moments. IgG antibody titers are expressed as geometric means with 95% confidence interval (95% CI). Data regarding IgG titers were analyzed at Napierian logarithm (ln)-transformed basis using Serology parameters were compared between groups (ARD and CG) and timepoints (D69 and D210) using GEE with normal marginal distribution and gamma distribution, respectively, and identity binding function assuming first order autoregressive correlation matrix between moments. Results were followed by Bonferroni multiple comparisons to identify differences between groups and timepoints. All analyses were two-sided, without adjustments for multiple comparisons. *ARD* autoimmune rheumatic diseases, *CG* control group.

Additionally, ABA and RTX were also reported as harmful to other vaccines' immunogenicity[16]. With regard to anti-TNF, one-third of patients were in combination with MTX, and this latter drug may partially account for this finding since it was reported to induce a reduced antibody response[3]. For NAb positivity, only ABA was associated with the absence of these antibodies at D210 in ARD patients. Likewise, a significantly lower NAb activity level was reported in men and immunosuppressed health worker participants 6 months after receipt of the second dose[8].

The predominance of incident cases in the 38 days after the vaccine's first shot (including 10 days after the second dose) contrasts with the significant drop in infection and hospitalization in the study participants in the subsequent 40 days. This last phase coincided with the second peak of COVID-19 cases in São Paulo city (45% increase in the same time period)[17]. Interestingly, the emergence of Delta (B.1.617.2) variant in Sao Paulo in July[18] with a rapid spread in the following months did not lead to a parallel upsurge of COVID-19 breakthrough cases in our cohort which remained with a homogeneous distribution of cases throughout the study period, supporting the notion that Sinovac-CoronaVac may maintain its effectiveness during 6 months.

In conclusion, we provide data on the long-term Sinovac-CoronaVac immunogenicity in ARD patients, demonstrating a significant decrease in IgG and NAb levels 6 months after the 2nd dose without a corresponding rise in symptomatic COVID-19 cases in the same period. Male, PRED, and biological therapy were identified as the main contributing factors to the reduced durability of the vaccine-induced humoral response.

## Methods

### Study design and population

This work is within a phase 4 prospective longitudinal study (CoronavRheum clinicaltrials.gov #NCT04754698) conducted at a large tertiary hospital in Sao Paulo, Brazil. The protocol complies with all relevant ethical regulations, and it was approved by the National and Institutional Ethical Committee of Hospital das Clinicas HCFMUSP, Faculdade de Medicina, University of Sao Paulo, Sao Paulo, Brazil (CAAE: 42566621.0.0000.0068). For this study, only ARD patients followed at our Outpatient Rheumatology Clinics were included, irrespective of their SARS-CoV-2 serologic status before vaccination. The patients were diagnosed according to international guidelines and disease classification criteria, and the population included were rheumatoid arthritis[19], systemic erythematosus lupus[20], axial ankylosing spondylitis[21], psoriatic arthritis[22], primary vasculitis[23,24], primary Sjogren's syndrome[25], systemic sclerosis[26], idiopathic Inflammatory myopathies[27], and primary antiphospholipid syndrome[28]. After the selection of the ARD group, hospital services workers, health professionals, and hospital administrative service employees or their relatives, without ARD or immunosuppressive therapy, were selected as a

healthy control group (CG). All participants were ≥18 years old, and the controls were balanced with patients by sex and age (up to 5 years differences) at the entry (1 control: 4 patients) using an in-house program run on Excel (Microsoft 2018) for random selection of individuals in each group. All ARD patients and CG received the inactivated Sinovac-CoronaVac vaccine (Sinovac Life Sciences, Beijing, China, batch #20200412) first dose between February 9 and 17, 2021, and the second dose between March 9 and 17, 2021 (28 days apart). No strategy for immunogenicity improvement was applied. The exclusion criteria at vaccination included previous vaccination with any SARS-CoV-2 vaccine, previous history of anaphylactic events after vaccine administration, history of previous vaccination with live virus up to 4 weeks, or with inactivated virus vaccine up to 2 weeks, COVID-19-related symptoms up to 4 weeks before entry, acute febrile illness, Guillain-Barré syndrome, decompensated heart failure, demyelinating disease, people living with HIV/AIDS (PLWHA), history of blood products administration up to 6 months before the study start. Hospitalized patients or controls were also excluded. The full flow diagram with all exclusions is presented in Fig. 1. The final study groups for immunogenicity analyses consisted of 828 ARD patients (mean age 49.8 ± 12.3 years, 75.5% female sex) and 207 healthy controls (mean age 49.7 ± 12.3 years, 75.4% female sex). Blood collections for this study were performed 6 weeks after the second dose (April 19, 2021, D69) and 6 months after the second dose (September 18, 2021, D210) for ARD and CG. All participants signed the written informed consent prior to the vaccination and blood collection. Compensation for participation was not provided. Study protocol with data collection and outcomes information is available in the Supplementary Information.

A rigorous follow-up of incident cases was performed for all participants (1115 ARD patients, mean age 49.7 ± 12.7 years, 76.7% female sex and 223 CG, mean age 49.6 ± 12.5 years, 76.7% female sex) with COVID-19 symptoms from vaccine first dose to 6 months after the second dose (D210). Four in-person visits (vaccine first dose, vaccine second dose, D69, and D210) were performed, with careful checking of a standardized diary regarding COVID-19 history at each visit. All ARD patients and controls were instructed to communicate any manifestation associated with COVID-19 through telephone, smartphone instant messaging, or email. A medical team was available for 24 h to provide a proper follow-up. Suspicious cases of COVID-19, even if mild, were instructed to seek medical care near the residence and to come to our tertiary hospital to undergo a RT-PCR test for SARS-Cov-2. Participants who were unable to come to our center were instructed to go to an independent laboratory near their homes. COVID-19 incident cases were followed from the first vaccine dose to 10 days after the second dose (T1) and thereafter, T2, for the following 170 days (from D40 to D210).

These participants were tested using a real-time reverse transcriptase-polymerase chain reaction (RT-PCR) test for SARS-CoV-2 by naso- and oropharyngeal swabs.

**Table 3 | Univariate analyses of characteristics of autoimmune rheumatic diseases (ARD) patients without and with ser-opositivity (SP) for anti-SARS-CoV-2 S1/S2 IgG antibodies and without and with neutralizing antibodies (NAbs 6 months after second dose of Sinovac-CoronaVac vaccination (D210)**

| | Patients without SP (anti-S1/S2 IgG) (n = 396) | Patients with SP (anti-S1/S2 IgG) (n = 560) | p-value | Patients without Nab (n = 599) | Patients with Nab (n = 357) | p-value |
|---|---|---|---|---|---|---|
| **Demographic data** | | | | | | |
| Current age, years | 53 (42–63) | 50 (39–60) | 0.001 | 52 (40–62) | 51 (40–60) | 0.382 |
| Current age ≥60 years | 128 (32.3) | 153 (27.3) | 0.094 | 182 (30.4) | 99 (27.7) | 0.384 |
| Male sex | 115 (29.0) | 116 (20.7) | 0.007 | 155 (25.9) | 76 (21.3) | 0.219 |
| Caucasian race | 214 (54.0) | 281 (50.2) | 0.239 | 309 (51.6) | 186 (52.1) | 0.878 |
| **ARD** | | | | | | |
| RA | 140 (35.4) | 136 (24.3) | <0.001 | 184 (30.7) | 92 (25.8) | 0.102 |
| SpA | 52 (13.1) | 69 (12.3) | 0.711 | 81 (13.5) | 40 (11.2) | 0.297 |
| PsA | 41 (10.4) | 55 (9.8) | 0.787 | 63 (10.5) | 33 (9.2) | 0.526 |
| SLE | 80 (20.2) | 158 (28.2) | 0.005 | 134 (22.4) | 104 (29.1) | 0.019 |
| Systemic vasculitis | 27 (6.8) | 34 (6.1) | 0.642 | 39 (6.5) | 22 (6.2) | 0.818 |
| IIM | 18 (4.5) | 25 (4.5) | 0.952 | 23 (3.8) | 20 (5.6) | 0.203 |
| SSc | 10 (2.5) | 31 (5.5) | 0.024 | 23 (3.8) | 18 (5.0) | 0.375 |
| SS | 12 (3.0) | 29 (5.2) | 0.106 | 25 (4.2) | 16 (4.5) | 0.820 |
| PAPS | 16 (4.0) | 23 (4.1) | 0.959 | 27 (4.5) | 12 (3.4) | 0.715 |
| **Current therapies** | | | | | | |
| Hydroxychloroquine | 93 (23.5) | 185 (33.0) | 0.001 | 167 (27.9) | 111 (31.1) | 0.290 |
| Sulfasalazine | 42 (10.6) | 44 (7.9) | 0.143 | 63 (10.5) | 23 (6.4) | 0.033 |
| Prednisone | 173 (43.7) | 174 (31.1) | <0.001 | 225 (37.6) | 122 (34.2) | 0.292 |
| Prednisone dose | 5 (5–10) | 7.5 (5–10) | 0.043 | 5 (5–10) | 7.5 (5–10) | 0.131 |
| Prednisone >10 mg/day | 22 (12.7) | 32 (18.4) | 0.145 | 29 (12.9) | 25 (20.5) | 0.062 |
| Prednisone >20 mg/day | 4 (2.3) | 6 (3.4) | 0.750 | 6 (2.7) | 4 (3.3) | 0.746 |
| Immunosuppressive | 255 (64.4) | 338 (60.4) | 0.205 | 368 (61.4) | 225 (63.0) | 0.624 |
| Methotrexate | 105 (26.5) | 144 (25.7) | 0.781 | 153 (25.5) | 96 (26.9) | 0.646 |
| Leflunomide | 62 (15.7) | 71 (12.7) | 0.190 | 86 (14.4) | 47 (13.2) | 0.606 |
| Mycophenolate mofetil | 53 (13.4) | 72 (12.9) | 0.812 | 73 (12.2) | 52 (14.6) | 0.291 |
| Azathioprine | 38 (9.6) | 63 (11.2) | 0.412 | 60 (10.0) | 41 (11.5) | 0.475 |
| Tofacitinib | 9 (2.3) | 14 (2.5) | 0.821 | 14 (2.3) | 9 (2.5) | 0.858 |
| Cyclophosphamide | 1 (0.3) | 6 (1.1) | 0.250 | 5 (0.8) | 2 (0.6) | 0.630 |
| Biologic agent | 179 (45.2) | 165 (29.5) | <0.001 | 231 (38.6) | 113 (31.7) | 0.031 |
| Anti-TNF | 76 (19.2) | 74 (13.2) | 0.012 | 102 (17.1) | 48 (13.4) | 0.141 |
| Abatacept | 39 (9.8) | 15 (2.7) | <0.001 | 42 (7.0) | 12 (3.4) | 0.018 |
| Tocilizumab | 20 (5.1) | 32 (5.7) | 0.656 | 28 (4.7) | 24 (6.7) | 0.177 |
| Belimumab | 16 (4.0) | 15 (2.7) | 0.242 | 19 (3.2) | 12 (3.4) | 0.873 |
| Rituximab | 12 (3.0) | 6 (1.1) | 0.028 | 14 (2.3) | 4 (1.1) | 0.181 |

Results are expressed in median (interquartile range) and n (%). Continuous data were compared using Mann–Whitney test, and categorical variables with the chi-square or Fisher's exact tests, as appropriate, always as two-sided analyses, without adjustments for multiple comparisons. SP—seropositivity (IgG titer ≥ 15 AU/ml) for anti-SARS-CoV-2 S1/S2 IgG antibodies after vaccination (Indirect ELISA, LIAISON® SARS-CoV-2 S1/S2 IgG, DiaSorin, Italy). Positivity for NAb was defined as a neutralizing activity ≥30% (cPass sVNT Kit, GenScript, Piscataway, USA).
*RA* rheumatoid arthritis, *SpA* spondyloarthritis, *PA* psoriatic arthritis, *SLE* systemic lupus erythematosus, *IIM* idiopathic inflammatory myopathy, *SSc* systemic sclerosis, *SS* Sjögren's syndrome, *PAPS* primary antiphospholipid syndrome.

## Primary and secondary outcomes

The primary outcome was humoral immunogenicity assessed by the presence of anti-S1/S2 SARS-CoV-2 IgG 6 months after the second vaccine dose (D210) compared to the same parameter at D69 (6 weeks after second vaccine dose) for decay evaluation. Secondary outcomes were: the presence of neutralizing antibodies (NAb), percent inhibition by NAb, and anti-S1/S2 SARS-CoV-2 IgG GMT. Incident cases confirmed by RT-PCR for SARS-CoV-2 infection were also a secondary outcome.

## Serologic assays

Serology was assessed for all participants prior to the first vaccine dose, at D69 (interval from second dose to serum collection = 6 weeks) and 6 months after the second dose (D210). The

blood samples were collected, centrifuged, and stored at −80 °C until analysis. The serologic assay consisted of the measurement of the total IgG antibodies against the SARS-CoV-2 S1 and S2 proteins performed by chemiluminescent immunoassay on the ETI-MAX-3000 equipment (DiaSorin, Italy) using the Indirect ELISA, LIAISON® SARS-CoV-2 S1/S2 IgG kit Cat# 311450 and Cat# 311451 (DiaSorin, Italy), and the measurement of the circulating NAb against SARS-CoV-2 using the SARS-CoV-2 sVNT Kit, Cat# L00847-A (GenScript, Piscataway, NJ, USA). The two assays were performed following the manufacturer's instructions. Samples with 15.0 UA/mL or more for total anti-S1/S2 IgG and with 30% or more inhibition for neutralizing assay, were considered seropositive according to the manufacturer's guide[29,30]. Furthermore, quantitative results were reported, attributing the value of 1.9 UA/mL (half of the lower limit of quantification 3.8

**Table 4 | Multivariate analyses of characteristics of auto-immune rheumatic diseases (ARD) patients without and with seropositivity (SP) for anti-SARS-CoV-2 S1/S2 IgG antibodies and without and with neutralizing antibodies (NAb) 6 months after second dose of Sinovac-CoronaVac vaccination (D210)**

| | Anti-S1/S2 IgG seropositivity | | | NAb positivity | | |
|---|---|---|---|---|---|---|
| | OR | 95% CI | p-value | OR | 95% CI | p-value |
| **Demographic data** | | | | | | |
| Current age, years | 0.99 | 0.98–1.0 | 0.053 | — | — | — |
| Male sex | 0.56 | 0.40–0.79 | <0.001 | — | — | — |
| **ARD** | | | | | | |
| RA | 0.92 | 0.62–1.36 | 0.667 | 0.98 | 0.70–1.37 | 0.911 |
| SLE | 0.97 | 0.59–1.57 | 0.886 | 1.21 | 0.87–1.69 | 0.265 |
| SSc | 1.57 | 0.73–3.38 | 0.248 | — | — | — |
| **Current therapies** | | | | | | |
| Hydroxychloroquine | 1.25 | 0.83–1.89 | 0.282 | — | — | — |
| Sulfasalazine | 1.03 | 0.64–1.67 | 0.891 | 0.65 | 0.39–1.09 | 0.101 |
| Prednisone | 0.56 | 0.41–0.76 | <0.001 | — | — | — |
| Leflunomide | 1.04 | 0.69–1.57 | 0.850 | — | — | — |
| **Biologic agent** | | | | | | |
| Anti-TNF | 0.66 | 0.45–0.96 | 0.031 | 0.80 | 0.54–1.19 | 0.271 |
| Abatacept | 0.29 | 0.15–0.56 | <0.001 | 0.48 | 0.24–0.97 | 0.041 |
| Tocilizumab | — | — | — | 1.55 | 0.83–2.91 | 0.174 |
| Rituximab | 0.32 | 0.11–0.90 | 0.031 | 0.45 | 0.15–1.38 | 0.161 |

Results are expressed in OR (95% CI) regarding the positivity for anti-S1/S2 IgG and for NAb. Adjusted analyses included the factors with p > 0.20 at unadjusted analyses, depicted in Table 3, using multiple logistic regression models.SP—seropositivity (IgG titer ≥ 15 AU/ml) for anti-SARS-CoV-2 S1/S2 IgG antibodies after vaccination (Indirect ELISA, LIAISON® SARS-CoV-2 S1/S2 IgG, DiaSorin, Italy). Positivity for NAb was defined as a neutralizing activity ≥30% (cPass sVNT Kit, GenScript, Piscataway, USA).
*RA* rheumatic arthritis, *SpA* spondyloarthritis, *PA* psoriatic arthritis, *SLE* systemic lupus erythematosus, *IIM* idiopathic inflammatory myopathy, *SSc* systemic sclerosis, *SS* Sjögren's syndrome, *PAPS* primary antiphospholipid syndrome.

UA/mL) to undetectable levels (<3.8 UA/mL) of total IgG and 15% (half of the lower limit of quantification 30%) to undetectable levels (<30%) of neutralizing antibodies.

## RT-PCR for SARS-CoV-2 analysis

Naso- and oropharyngeal swabs were obtained whenever the participants reported COVID-19-related symptoms, and SARS-CoV-2 RT-PCR was performed using the SuperScriptTM III Platinum® One-Step Quantitative RT-PCR System Cat#11732-088 (Invitrogen, Carlsbad, CA, USA). A m2000sp equipment (Abbott, Des Plaines, IL, USA) was for RNA extraction and a m2000rt (Abbott, Des Plaines, IL, USA) was for RNA amplification and detection. The assay was carried out according to the protocol reported by Corman and colleagues as previously reported assay[31].

## Statistical analysis and reproducibility

Participants originated from a large phase 4 controlled prospective study of immunogenicity and safety of two doses of Sinovac-CoronaVac vaccine, with its proper sample size calculation[13]. No statistical method was used to predetermine the sample size in the current research. Demographical data were presented as number (percentage) for categorical variables and as mean ± standard deviation, and median (interquartile ranges) for continuous variables. The comparison between ARD and CG was performed by the chi-square or Fisher's exact tests, as appropriate for categorical variables, and by Mann–Whitney test for continuous variables.

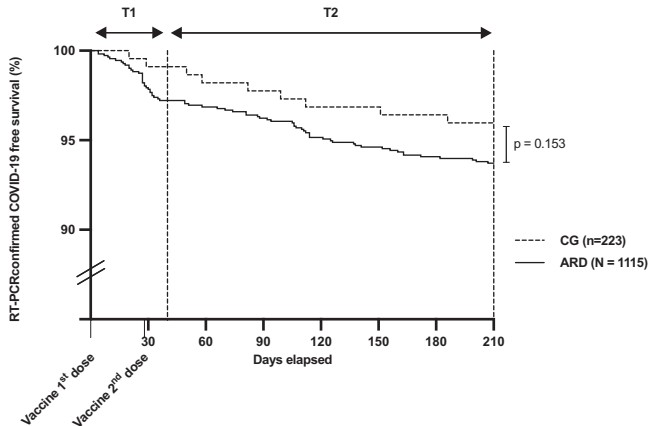

**Fig. 2 | Kaplan–Meier curves of COVID-19 incident cases.** The figure depicts real-time polymerase chain reaction (RT-PCR) confirmed COVID-19 cases in patients with autoimmune rheumatic disease (ARD) and controls (CG). All participants were followed from the day of inclusion, which is the day of first dose vaccination with CoronaVac vaccine (DO) up to 10 days after the second dose vaccination (T1). The second period of follow-up (T2) ranged from T1 to 6 months after the second dose. Kaplan–Meier curves were compared with long-rank test and if *p*-value was <0.05 then the difference was considered significant.

**Immunogenicity assessment.** ARD patients and the controls were balanced with patients by sex and age (up to 5 years differences) at the entry (1 control: 4 patients) using an in-house program run on Excel (Microsoft 2018) for random selection of individuals in each group. Seropositivity (SP) rates of anti-S1/S2 IgG and NAb were presented as numbers (percentage) of positive samples and were compared between groups (ARD and CG) and between timepoints (D69 vs. D210) using generalized estimating equations (GEE) with binomial distribution and logit link function, assuming autoregressive correlation matrix between moments. IgG antibody titers are expressed as geometric means with 95% confidence interval (95% CI). Total anti-S1/S2 SARS-CoV-2 IgG was expressed as geometric mean titers (GMT) and 95% confidence intervals, and it was compared using Generalized Estimating Equations (GEE) with two factors [2 groups (ARD and CG) at two timepoints (D69 vs. D210)] followed by Bonferroni's multiple comparisons in neperian logarithm (ln)-transformed data. Percent inhibition by NAb was expressed as medians (interquartile range) of the percentage of inhibition and was compared using Generalized Estimating Equations (GEE) with two factors [2 groups (ARD and CG) at two timepoints (D69 vs. D210)] followed by Bonferroni's multiple comparisons. Multivariate logistic regression analyses were performed using as dependent variables SC or NAb positivity at D210 and as independent variables those with *p* < 0.2 in each univariate analysis (for IgG SC: current age, male sex, rheumatoid arthritis, systemic lupus erythematosus, systemic sclerosis, hydroxychloroquine, sulfasalazine, prednisone, leflunomide, anti-TNF, abatacepte, and rituximabe use; for Nab: rheumatoid arthritis, systemic lupus erythematosus, sulfasalazine, anti-TNF, abatacept, tocilizumab, and rituximab use).

**COVID-19 incident cases.** Confirmed COVID-19 cases and hospitalization incident-density data (along with 95% confidence intervals) were estimated using Poisson distribution and test-based methods and compared on ARD patients at T1 (from the 1st dose to 10 days after the second dose) and T2 (180 days after the second dose), with significance related to incident differences. Survival analysis of ARD and CG subjects free of confirmed COVID-19 cases was performed using Kaplan–Meier curves and compared with the long-rank test. Statistical significance was defined as *p* < 0.05.

Most statistical analyses were performed using Statistical Package for the Social Sciences, version 20.0 (IBM-SPSS for Windows. 20.0. Chicago, IL, USA). For incident-density data, MedCalc version 20.015 was used.

## Reporting summary

Further information on research design is available in the Nature Research Reporting Summary linked to this article.

## Data availability

Raw data for each figure or table have been deposited as a Source Data file. All data used in these analyses can be found at (https://figshare.com/s/b25763a08f5d782f50ce). The raw personal data are protected and are not available due to data privacy laws. Source data are provided with this paper.

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

## Acknowledgements

This study was sponsored by grants from Fundação de Amparo à Pesquisa do Estado de São Paulo (FAPESP) (#2015/03756–4 to C.A.S., S.G.P., N.E.A., and E.B.; #2019/17272-0 to L.V.K.K.; #2018/09937-9 to V.O.A.M., #2020/09367-8 to L.E.B.V.), Conselho Nacional de Desenvolvimento Científico e Tecnológico (#304984/2020-5 to C.A.S.; #305556/2017-7 to R.M.R.P.; #303379/2018-9 and to S.K.S. #305242/2019-9 to E.B.), and B3-Bolsa de Valores do Brasil and Instituto Todos pela Saúde (ITPS 01/2021, C1313 to C.A.S., S.G.P., N.E.A., and E.B.). Instituto Butantan supplied the study product and had no other role in the trial. We thank the

volunteers for participating in all in-person visits and for handling the biological material. We also thank the contribution of the Central Laboratory Division, Registry Division, Security Division, IT Division, Superintendence, Pharmacy Division, and Vaccination Center for their technical support.

## Author contributions

C.A.S., A.C.M.R., L.V.K.K., N.E.A., C.G.S., E.F.N.Y., S.G.P., C.G.S., E.G.K., and E.B. conceived and designed the study and participated in data collection and analysis, supervised clinical data management, writing of the manuscript, and revision of the manuscript. S.R.G.F. organized and supervised blood collection and vaccination. A.J.S.D. supervised sera processing, SARS-CoV-2-specific antibody ELISAs/neutralization assays, and SARS-CoV-2 RT-PCR. C.A.S., A.C.M.R., L.V.K.K., N.E.A., C.G.S., E.F.N.Y., S.G.P., C.G.S., E.B., S.R.G.F., R.M.R.P., S.K.S., A.S.R.H., E.F.B., F.H.C.S., L.K.N.G., R.M., K.R.B., D.S.D., A.Y.S., D.C.O.A., L.P.C.S., R.F., P.D.S.B., A.P.L.A., J.C.B.M., C.G.S., H.A.M.G., H.C.S., V.A.O.M., L.E.B.V., R.S.N., L.P.S., C.S.R.A., M.S.R.S., and D.M.N.F. collected epidemiological and clinical data and assisted with the identification of SARS-CoV-2 infection and follow-up of patients. M.H.L. organized and supervised the vaccination protocol. A.J.S.D. supervised and performed the assays for measurement of anti-SARS-CoV-2 antibodies. All authors helped to edit the manuscript.

## Competing interests

The authors declare no competing interests.

## Additional information

**Clovis A. Silva[1], Ana C. Medeiros-Ribeiro [2], Leonard V. K. Kupa[2], Emily F. N. Yuki [2], Sandra G. Pasoto[2], Carla G. S. Saad [2], Solange R. G. Fusco[2], Rosa M. R. Pereira[2], Samuel K. Shinjo[2], Ari S. R. Halpern[2], Eduardo F. Borba[2], Fernando H. C. Souza[2], Lissiane K. N. Guedes[2], Renata Miossi[2], Karina R. Bonfiglioli[2], Diogo S. Domiciano[2], Andrea Y. Shimabuco[2], Danieli C. O. Andrade[2], Luciana P. C. Seguro[2], Ricardo Fuller[2], Percival D. Sampaio-Barros[2], Ana P. L. Assad[2], Julio C. B. Moraes[2], Claudia Goldenstein-Schainberg [2], Henrique A. M. Giardini[2], Henrique C. Silva[2], Victor A. O. Martins[2], Lorena E. B. Villamarin [2], Renata S. Novellino [2], Lucas P. Sales[2], Carlo S. R. Araújo[2], Matheus S. R. Silva[2], Dilson M. N. Filho[2], Marta H. Lopes[3], Alberto J. S. Duarte[4], Esper G. Kallas[3], Nadia E. Aikawa [1] & Eloisa Bonfa [2] ✉**

[1]Pediatric Rheumatology Unit, Instituto da Criança e do Adolescente, Hospital das Clinicas HCFMUSP, Faculdade de Medicina, Universidade de Sao Paulo, Sao Paulo, Brazil. [2]Rheumatology Division, Hospital das Clinicas HCFMUSP, Faculdade de Medicina, Universidade de Sao Paulo, Sao Paulo, Brazil. [3]Infectious Disease Department, Hospital das Clinicas HCFMUSP, Faculdade de Medicina, Universidade de Sao Paulo, Sao Paulo, Brazil. [4]Central Laboratory Division, Hospital das Clinicas HCFMUSP, Faculdade de Medicina, Universidade de Sao Paulo, Sao Paulo, Brazil. ✉e-mail: eloisa.bonfa@hc.fm.usp.br

