## [Peer Review File · Nature Communications]

Immunogenicity decay and incident cases six months after
Sinovac-CoronaVac vaccine in autoimmune rheumatic
diseases patientsREVIEWER COMMENTS

Reviewer #1 (Remarks to the Author):

This is a study of great interest to the community in the face of the need to have enough vaccines to vaccinate all the people of the world and also to learn more about the immune response capacity of the vaccines and their persistence or durability in the time in order to design global vaccination strategies. So, I congratulate to the authors for this work

The study's objectives are assessing the immunogenicity of Sinovac-Coronavac vaccine six months after the administration of second dose. The study wants to analyse the breakthrough infections and determine the risk factors associate with loss of immunity.

According with the authors the key findings of the study are that after 6-months of Sinovac-CoronaVac 2nd dose immunogenicity of ARD patients was reduced, particularly in males and those under prednisone or some biological therapies, especially anti-TNF, Abatacept and rituximab without a concomitant rise in COVID-19 cases.

The study demonstrates technical rigor. I'll just comment on some aspects of this paper with the intention that these comments will help you to improve it and so it may be easier to read.

Major comments

1. Group control: I miss an explanation of the characteristics of the participants in this group. For example: are they a healthy people or not? Are they vaccinated at the same time of the participants in the intervention group?

2. I also miss an explanation of the follow up of participants to detect incident cases. What is the procedure used? Regular visits or calls or there are a passive surveillance system or review of the health records? Are you sure that you have detected all the incident cases?

3. In these lines you write:

203 . We further demonstrated that the same phenomena, although

204 less intense, occurred for the Sinovac-CoronoVac with a more expressive decrease in 205 controls (51%) than in ARD patients (38%) at 6 months post second dose.

How explain that the the decrease is greater in the control group?

Minor comments

1. In line 102 you write: 62.0% (n=513) immunosuppressive drugs, 36.5% (n=302) prednisone and 35.7% 103 (n=296) biologic therapy.

I think it would be the interest that you specify which drugs have been considered within the group of immunosuppressive drugs and which within the group of biologics and whether the dose of prednisone has been taken into account in the analysis? Although it is specified in the tables I think it should be commented in the text or referenced the table

2. I think that the description of the characteristic of the patients who die doesn't add value to the paper and this isn't include as an objective of this study.

3. In the discussion there are some sentences that was difficult to understand. I think a review of the redaction can improve it.

In line 183 . The use of established classification criteria for 184 each disease allowed a more accurate definition of the impact of different conditions 185 and their therapy. What classification?

When the authors use this classification? Do you refer that you use this classification for the diagnosis of patients included in this study?....

In line 201 of approximately 20% for patients and controls. We also confirmed the observation of 202 a robust (≥ 4 times) decrease in SARS-CoV-2 mRNA vaccine IgG levels after 6-months in health care workers⁸ You confirmed this finding with the results of your study or is a finding for others studies?

Anna Vilella

Reviewer #2 (Remarks to the Author):

This paper describes the effect of immunosuppression or underlying disease on maintenance of immunity following two doses of Sinovac-CoronaVac. The inclusion/exclusion of participants is well described.

Abstract and introduction

Some of the sentence structure in the abstract and introduction makes the writing difficult to follow at times. I think some points are a bit lost as a result.

Line 55: change doses to dose

Line 56-59: poor sentence structure, hard to follow

line 61: change "in 6 months" to "by 6 months".

Line 68: Significant reduction in levels of what? Antibodies or virus positive cases?

Line 69-71: poor sentence, hard to follow

Results

line 108: change to "reduced by 23.8%"

line 116: change to "declined by 41%"

line 120: change sentence from "The decrease in NAb activity [$1 - \ln(\text{NAb activity}_{D210}/\text{NAb activity}_{D69})$] in positive individuals at D69 was significantly higher in ARD compared to CG [54% (95%CI 51-57%) vs. 47% (95%CI 41-52%); $p=0.024$]."

This would read better written as " There was a significantly greater reduction in NAb titres in those with positive NAb titres in the ARD vs the CG [54% (95%CI 51-57%) vs. 47% (95%CI 41-52%); $p=0.024$]."

Discussion

Line 220-224: This sentence is very hard to follow, meaning is lost.

It would have been interesting if some discussion around the effects of anti-TNF, prednisone and rituximab were discussed considering the detrimental effect these therapies have on the generation of IgG responses. Also, some discussion as to why quantity of antibody appears to have been affected more than function(NAb)?

Reviewer #3 (Remarks to the Author):

I read with interest the paper entitled "Anti-SARS-CoV-2 immunogenicity decay and incident cases six months after Sinovac-CoronaVac inactivated vaccine in autoimmune rheumatic diseases patients: phase 4 prospective trial" in which the authors reported the decay in immunogenicity and the incidence of COVID-19 cases six months after the 2nd dose of Sinovac-CoronaVac.

Additionally, the authors evaluated risk factors for the reduced 6-month duration of anti-SARS-CoV-2 immunogenicity, finding that prednisone, immunosuppressive drugs, and male sex play a role in immunogenicity decay.

↵

MAJOR COMMENTS

- Dataset

The legend is missing about the meaning of coloured lines, some columns are badly formatted, it is not clear which patients have been selected for analysis. Therefore, the dataset is unfeasible for the reviewer.

- Methods section

Exclusion criteria

The anti-S1/S2 IgG assessment at baseline, prior to vaccination, is not clearly evident in the paper (methods and results section).

Serologic assays

The authors reported that they have evaluate the presence of neutralizing antibodies (NAb) using SARS-CoV-2 Surrogate Virus Neutralization Test (sVNT) Kit.

The sVNT kit detects and measures anti-RBD IgG with ACE2 protein attached to the plate and HRP-labeled RBD used for detection. This kit only mimics the virus neutralization process.

Lustig and colleagues reported a significant correlation between anti-RBD and neutralization titers, evaluated with SARS-CoV-2 pseudo-virus neutralization assay.

Considering the results of Lustig and colleagues anti-RBD might be considered a correlate of neutralization, but the authors cannot conclude that they have evaluate neutralizing activity.

Lustig Y, Sapir E, Regev-Yochay G, et al. BNT162b2 COVID-19 vaccine and correlates of humoral immune responses and dynamics: a prospective, single-centre, longitudinal cohort study in health-care workers. *Lancet Respir Med.* 2021;9(9):999-1009. doi:10.1016/S2213-2600(21)00220-4

• Results section

The authors should expand the methodology and report the elements included in the univariate and multivariate analysis, a table could be useful.

Patients with history of SARS-CoV-2 infection before vaccination should be analyzed as a sub-group.

The use of prednisone was significantly associated with the absence of anti-S1/S2 IgG six months after vaccination in ARD patients. It would be useful to know the median daily dose of prednisone; although, according to the dataset, the dose of prednisone is missing in 1236 of 1687 people.

Unexpectedly, the use of anti-TNF drugs resulted significantly associated with absence of anti-S1/S2 IgG six months after vaccination in ARD patients. In previously reported real word studies, anti-TNF did not influence the immunogenicity. Did the authors consider the role of the combination with MTX?

The immunogenicity status of ARD patients who were infected with SARS-CoV-2 during the follow-up and who died from COVID-19 should be reported.

• Limitations section

I believe the authors should rephrase this sentence “An important limitation of the present study is the non-assessment of cellular immunity in this population, but neutralizing antibodies evaluated herein were reported to be associated with protective immunity”. In fact, the detection of anti-RBP cannot exceed the limit of the non-evaluation of the T-cell response.

MINOR COMMENTS

- The following sentence (Discussion section) is not well balanced.

Our data supports the notion that Sinovac-CoronaVac may sustain a 6-month effectiveness against Delta strain.

- Please, add abbreviations and correct the typos.

- The referee has some problems with the way the data are presented in the diagram (Supplemental Material 1), the authors should clarify the ARD group and the control group in the diagram and better format it.

- Supplementary table 1: please change Caucasian race with Caucasian ethnicity

- D69 does not correspond to number of the days in 6 weeks, the same for D210 for 6 months

Reviewer #4 (Remarks to the Author):

The paper entitled “Anti-SARS-CoV-2 immunogenicity decay and incident cases six months after Sinovac- CoronaVac inactivated vaccine in autoimmune rheumatic diseases patients: phase 4 prospective trial” provides data about 6 months immunogenicity of the anti-SARS-CoV2 inactivate vaccine in patients with autoimmune rheumatic diseases. The study shows a significant difference between patients and control in the percentage of seropositive subjects (both after 6 weeks and 6 months from the vaccination) and 20% decline in the percentage of positive subject after 6 months in both groups. The authors found a decline in the titer of IgG anti-SARS CoV2 at day 210 post vaccine (compared to day 69) that was significantly greater in control subjects (51% vs 38%). On the contrary, the prevalence and the titer of neutralizing antibodies decrease significantly more in patients with autoimmune rheumatic disease compared with controls. Finally, the authors extended the efficacy data to the 6 months observation showing a decline of the infection from 10 days to 180 days after the vaccination completion.

The multivariate analysis found male gender, glucocorticoids and biological drugs as the factors associated to a greater decrease in the vaccine immunogenicity.

The paper is interesting and original since no previous studies reported data on the long-term immunogenicity and efficacy of Sinovac- CoronaVac.

I have only minor points to consider:

- In the supplementary table 2 the authors divided the patients according to daily prednisone dose (<10, >10, >20 mg/day). I wonder if different prednisone doses have different impact on immunogenicity: did the authors include that in the multivariate analysis?
- In the Discussion section, I suggest to better point out that Nab account for the humoral response, but the cellular response also contribute to vaccine efficacy.
- Do the authors have any explanation about the lower reduction of IgG titer in patients compared to control?

January, 25th 2022

REVIEWER COMMENTS

Reviewer #1 (Remarks to the Author):

This is a study of great interest to the community in the face of the need to have enough vaccines to vaccinate all the people of the world and also to learn more about the immune response capacity of the vaccines and their persistence or durability in the time in order to design global vaccination strategies. So, I congratulate to the authors for this work

The study's objectives are assessing the immunogenicity of Sinovac-Coronavac vaccine six months after the administration of second dose. The study wants to analyse the breakthrough infections and determine the risk factors associate with loss of immunity.

According with the authors the key findings of the study are that after 6-months of Sinovac-CoronaVac 2nd dose immunogenicity of ARD patients was reduced, particularly in males and those under prednisone or some biological therapies, especially anti-TNF, Abatacept and rituximab without a concomitant rise in COVID-19 cases.

The study demonstrates technical rigor. I'll just comment on some aspects of this paper with the intention that these comments will help you to improve it and so it may be easier to read.

Major comments

1. Group control: I miss an explanation of the characteristics of the participants in this group. For example: are they a healthy people or not? Are they vaccinated at the same time of the participants in the intervention group?

Full details on the Control group characteristics and recruitment is now provided on Methods section (Page 1, lines 13-19).

ARD patients and CG were vaccinated and collected blood samples at the same time and it was now specified in the Methods section (Page 2, lines 38-39)

2. I also miss an explanation of the follow up of participants to detect incident cases. What is the procedure used? Regular visits or calls or there are a passive surveillance system or review of the health records? Are you sure that you have detected all the incident cases?

Follow up for incident cases is clarified in detail, as requested. Four in-person visits (vaccine first dose, vaccine second dose, D69 and D210) were performed, with a careful checking of a standardized diary regarding COVID-19 history at each visit. A close surveillance was performed by our medical team, which was available for 24-hours by telephone, smartphone instant messaging or email to provide a proper follow-up. Participants with suspicious cases of COVID-19, even if mild, were instructed to come to our tertiary hospital or to an independent laboratory near their home to undergo a RT-PCR test for SARS-Cov-2 (Methods

section, page 2, lines 41-53).

3. In these lines you write:

203 . We further demonstrated that the same phenomena, although
204 less intense, occurred for the Sinovac-CoronoVac with a more expressive
decrease in 205 controls (51%) than in ARD patients (38%) at 6 months post second
dose.

How explain that the the decrease is greater in the control group?

In fact, the magnitude of reduction in anti-S1/S2 IgG levels 6 months after vaccination was expressive in the control group, as previously demonstrated in healthy population (Levin EG 2021, Shrotri M 2021), and higher than ARD patients. It may be explained by the higher GMT peak level in the former group. In spite of that, the decrease in the IgG seropositivity rates was similar in both groups (24% in ARD and 20% in the control group). This point was included in the Discussion section (page 8, lines 283-286)

Minor comments

1. In line 102 you write: 62.0% (n=513) immunosuppressive drugs, 36.5% (n=302) prednisone and 35.7% 103 (n=296) biologic therapy.

I think it would be the interest that you specify which drugs have been considered within the group of immunosuppressive drugs and which within the group of biologics and whether the dose of prednisone has been taken into account in the analysis? Although it is specified in the tables I think it should be commented in the text or referenced the table

The specific drugs in the group of immunosuppressive and biologic drugs as well as the median dose of prednisone were now described in the text (Results section, page 4, lines 126-134).

As suggested, the dose of prednisone was also included in the analysis of anti-SARS-Cov-2 IgG and NAb positivity. Of note, seronegative patients for IgG at D210 were under a higher median prednisone dose [7.5 (5 - 10) vs. 5 (5 - 10) mg/day, p=0.043] than seropositive patients (Results section, page 5, line 161).

2. I think that the description of the characteristic of the patients who die doesn't add value to the paper and this isn't include as an objective of this study.

As suggested, the description of characteristics of deceased patients were excluded from the Results (page 5, lines 160-161)

3. In the discussion there are some sentences that was difficult to understand. I think a review of the redaction can improve it.

In line 183 . The use of established classification criteria for each disease allowed a more accurate definition of the impact of different conditions and their therapy. What classification? When the authors use this classification? Do you refer that you use this classification for the diagnosis of patients included in this study?....

We clarify now that we are referring to disease classification criteria for diagnosis of patients included in the study. (Methods section, page 1, line 9; Discussion section, page 7, lines 242-243)

In line 201 of approximately 20% for patients and controls. We also confirmed the observation of 202 a robust (≥ 4 times) decrease in SARS-CoV-2 mRNA vaccine IgG levels after 6-months in health care workers⁸ You confirmed this finding with the results of your study or is a finding for others studies?

The sentence was indeed not clear. We now rephrased it to explain that we are confirming the results of a previous study (page 7, lines 242-243)

Anna Vilella

Reviewer #2 (Remarks to the Author):

This paper describes the effect of immunosuppression or underlying disease on maintenance of immunity following two doses of Sinovac-CoronaVac. The inclusion/exclusion of participants is well described.

Abstract and introduction

Some of the sentence structure in the abstract and introduction makes the writing difficult to follow at times. I think some points are a bit lost as a result.

Line 55: change doses to dose

We have corrected the text, page 2, line 57

Line 56-59: poor sentence structure, hard to follow

We have corrected the text page 2, lines 58-62

line 61: change "in 6 months" to "by 6 months".

We have corrected the text page 2, line 63

Line 68: Significant reduction in levels of what? Antibodies or virus positive cases?

We have completed the text by adding "Antibodies titers" page 2, line 71

Line 69-71: poor sentence, hard to follow

We have corrected the text page 2, lines 73-76

Results

line 108: change to "reduced by 23.8%"

We have corrected the text page 5, line 141

line 116: change to "declined by 41%"

We have corrected the text page 5, line 149

line 120: change sentence from "The decrease in NAb activity [$1 - \ln(\text{NAb activity}_{D210}/\text{NAb activity}_{D69})$] in positive individuals at D69 was significantly higher in ARD compared to CG [54% (95%CI 51-57%) vs. 47% (95%CI 41-52%); $p=0.024$]."

This would read better written as " There was a significantly greater reduction in NAb titres in those with positive NAb titres in the ARD vs the CG [54% (95%CI 51-57%) vs. 47% (95%CI 41-52%); $p=0.024$]."

We have corrected the text page 5, lines 150-152

Discussion

Line 220-224: This sentence is very hard to follow, meaning is lost.

We have rephrased these sentences page X, line X

It would have been interesting if some discussion around the effects of anti-TNF, prednisone and rituximab were discussed considering the detrimental effect these therapies have on the generation of IgG responses.

We included a discussion about the deleterious effect of these drugs in IgG response (Discussion section, page 8, lines 296-320).

Also, some discussion as to why quantity of antibody appears to have been affected more than function (NAb)?

We rephrased the sentence to clarify that NAb positivity and NAb activity (percent inhibition) has a similar decrease in our study, contrasting with a more substantial reduction with mRNA vaccine (Discussion section, page 8, lines 287-289).

Reviewer #3 (Remarks to the Author):

I read with interest the paper entitled "Anti-SARS-CoV-2 immunogenicity decay and incident cases six months after Sinovac-CoronaVac inactivated vaccine in autoimmune rheumatic diseases patients: phase 4 prospective trial" in which the authors reported the decay in immunogenicity and the incidence of COVID-19 cases six months after the 2nd dose of Sinovac-CoronaVac.

Additionally, the authors evaluated risk factors for the reduced 6-month duration of anti-SARS-CoV-2 immunogenicity, finding that prednisone, immunosuppressive drugs, and male sex play a role in immunogenicity decay.

↵

MAJOR COMMENTS

- Dataset

The legend is missing about the meaning of coloured lines, some columns are badly formatted, it is not clear which patients have been selected for analysis. Therefore, the dataset is unfeasible for the reviewer.

We have made corrections in the Dataset to make it clearer for the reviewer.

- Methods section

Exclusion criteria

The anti-S1/S2 IgG assessment at baseline, prior to vaccination, is not clearly evident in the paper (methods and results section).

The primary outcome was anti-S1/S2 SARS-CoV-2 IgG behavior 6 months after

the 2nd vaccine dose (D210), and therefore serology status at baseline (D0) was not used as an inclusion or exclusion criteria.

But, as requested, this information is now provided in the Methods (Page 1, lines 7-8) and Results (Page 3, lines 138-140).

Serologic assays

The authors reported that they have evaluate the presence of neutralizing antibodies (NAb) using SARS-CoV-2 Surrogate Virus Neutralization Test (sVNT) Kit.

The sVNT kit detects and measures anti-RBD IgG with ACE2 protein attached to the plate and HRP-labeled RBD used for detection. This kit only mimics the virus neutralization process.

Lustig and colleagues reported a significant correlation between anti-RBD and neutralization titers, evaluated with SARS-CoV-2 pseudo-virus neutralization assay. Considering the results of Lustig and colleagues anti-RBD might be considered a correlate of neutralization, but the authors cannot conclude that they have evaluate neutralizing activity.

Lustig Y, Sapir E, Regev-Yochay G, et al. BNT162b2 COVID-19 vaccine and correlates of humoral immune responses and dynamics: a prospective, single-centre, longitudinal cohort study in health-care workers. *Lancet Respir Med.* 2021;9(9):999-1009. doi:10.1016/S2213-2600(21)00220-4

The term neutralizing activity was changed to percent inhibition by NAb to better suit the characteristics of the surrogate virus neutralization test used in the present study (Methods section, page 3, line 86, page 4, lines 111-112; Abstract section, page 2, line 30; Results section, page 5, line 151, page 8, line 289). The suggested Reference was also included in the Methods reference list (number #29).

- Results section

The authors should expand the methodology and report the elements included in the univariate and multivariate analysis, a table could be useful.

As suggested, elements included in univariate and multivariate analysis are better described in the Methods (page 4, lines 117-121) and Results sections (page 5, lines 185-186 and 195-196) . A new Table 3 was also included.

Patients with history of SARS-CoV-2 infection before vaccination should be analyzed as a sub-group.

A subanalysis of ARD patients with positive anti-SARS-CoV-2 serology before vaccination demonstrated a significant increase in anti-S1/S2 GMT from D0 to D69 ($p < 0.001$) with a subsequent decrease from D69 to D210 ($p < 0.001$) [74.7 (95%CI 64.4 - 86.6) vs 165.7 (95%I 143.2 - 191.7) vs 104.8 (95%CI 88.5 - 124.2), $p < 0.001$]. These data were presented in the Results section (Page 5, lines 153-156).

The use of prednisone was significantly associated with the absence of anti-S1/S2 IgG six months after vaccination in ARD patients. It would be useful to know the median daily dose of prednisone; although, according to the dataset, the dose of prednisone is missing in 1236 of 1687 people.

The median of daily dose of prednisone [5 (IQR 0.5 - 10) mg/day] was now provided in the Results section (page 4, lines 129-130), as suggested.

The Dataset has 2 columns concerning prednisone, one defining the use or not of prednisone and the other for entering the dose that can only be filled if the patient is taking this drug. Therefore, there is not actually missing data for ARD. With regard to the control group, none of them were under glucocorticoid and data were now completed in the dataset.

Unexpectedly, the use of anti-TNF drugs resulted significantly associated with absence of anti-S1/S2 IgG six months after vaccination in ARD patients. In previously reported real world studies, anti-TNF did not influence the immunogenicity. Did the authors consider the role of the combination with MTX?

In fact, one third of patients (28%) on anti-TNF were under combination with methotrexate. We have added a sentence to emphasize the relevant point raised by the reviewer. (Discussion section, page 9, lines 318-320)

The immunogenicity status of ARD patients who were infected with SARS-CoV-2 during the follow-up and who died from COVID-19 should be reported.

As suggested, the immunogenicity status of deceased ARD patients was now presented (Results section, page 7, lines 224-225).

- Limitations section

I believe the authors should rephrase this sentence “An important limitation of the present study is the non-assessment of cellular immunity in this population, but neutralizing antibodies evaluated herein were reported to be associated with protective immunity”. In fact, the detection of anti-RBP cannot exceed the limit of the non-evaluation of the T-cell response.

The sentence was rephrased to better characterize the limitation of anti-RBP as a measure of humoral response, not surpassing T-cell response evaluation (Results section, page 6, lines 273-275).

MINOR COMMENTS

- The following sentence (Discussion section) is not well balanced.

Our data supports the notion that Sinovac-CoronaVac may sustain a 6-month effectiveness against Delta strain.

We have reformulated the sentence as suggested (Discussion section, page 9, lines 328-332).

- Please, add abbreviations and correct the typos.

We have added abbreviations and corrected typo as recommended (Introduction section, Results section, Discussion section).

- The referee has some problems with the way the data are presented in the diagram (Supplemental Material 1), the authors should clarify the ARD group and the control group in the diagram and better format it.

The Flow chart was changed to better identify ARD group and control group in both Immunogenicity and COVID-19 incident cases analysis.

- Supplementary table 1: please change Caucasian race with Caucasian ethnicity
The term was changed, as suggested by the referee.

- D69 does not correspond to number of the days in 6 weeks, the same for D210 for 6 months

As now more clearly described in Methods, D69 corresponds to the time point of 6 weeks after the second vaccine dose to assure maximum vaccine response in all participants. And D210 corresponds to the evaluation made 6 months (180 days) after the second vaccine dose (Methods section, page 2, lines59 and 63, page 3, lines 75-77). We have reinforced that in the Results (Results section, page 5, lines137-140).

Reviewer #4 (Remarks to the Author):

The paper entitled “Anti-SARS-CoV-2 immunogenicity decay and incident cases six months after Sinovac- CoronaVac inactivated vaccine in autoimmune rheumatic diseases patients: phase 4 prospective trial” provides data about 6 months immunogenicity of the anti-SARS-CoV2 inactivate vaccine in patients with autoimmune rheumatic diseases. The study shows a significant difference between patients and control in the percentage of seropositive subjects (both after 6 weeks and 6 months from the vaccination) and 20% decline in the percentage of positive subject after 6 months in both groups. The authors found a decline in the titer of IgG anti-SARS CoV2 at day 210 post vaccine (compared to day 69) that was significantly greater in control subjects (51% vs 38%). On the contrary, the prevalence and the titer of neutralizing antibodies decrease significantly more in patients with autoimmune rheumatic disease compared with controls. Finally, the authors extended the efficacy data to the 6 months observation showing a decline of the infection from 10 days to 180 days after the vaccination completion.

The multivariate analysis found male gender, glucocorticoids and biological drugs as the factors associated to a greater decrease in the vaccine immunogenicity.

The paper is interesting and original since no previous studies reported data on the long-term immunogenicity and efficacy of Sinovac- CoronaVac.

I have only minor points to consider:

- In the supplementary table 2 the authors divided the patients according to daily prednisone dose (<10, >10, >20 mg/day). I wonder if different prednisone doses have different impact on immunogenicity: did the authors include that in the multivariate analysis?

Since we have already included prednisone use in the multivariate analysis, it was not possible to include its dose in the same analysis. However, we have performed the interesting suggestion made by the reviewer in the univariate analysis. There were no differences in immunogenicity according to the frequency of patients on >10 or >20 mg/day of prednisone (Table 2).

- In the Discussion section, I suggest to better point out that Nab account for the humoral response, but the cellular response also contribute to vaccine efficacy. **We added this important point in the Discussion section (page 8, lines 273-275) The sentence was rephrased to better characterize the limitation of anti-RBP as a measure of humoral response, not surpassing T-cell response evaluation.**

- Do the authors have any explanation about the lower reduction of IgG titer in patients compared to control?

In fact, the magnitude of reduction in anti-S1/S2 IgG levels 6 months after vaccination was expressive in the control group, as previously demonstrated in healthy population (Levin EG 2021, Shrotri M 2021), and higher than ARD patients. It may be explained by the higher GMT peak level in the former group. In spite of that, the decrease in the IgG seropositivity rates was similar in both groups (24% in ARD and 20% in the control group). This point was included in the Discussion section (page 8, lines 283-286)

REVIEWER COMMENTS

Reviewer #1 (Remarks to the Author):

I would like to congratulate the authors. The article has improved a lot after the review. I don't have more comments to do.

The study presents new results that are of interest to all those countries where these vaccines have been used and can help the vaccination strategies established.

Editorial Note: Reviewer #2 was unable to review on this occasion, so Reviewer #1 was asked to respond on their previous comments raised.

I've reviewed the responses to the concerns of reviewer 2.

The authors have responded correctly to the concerns of reviewer 2.

Reviewer #3 (Remarks to the Author):

The authors very carefully addressed the comments made, reviewed some confusing or unclear elements of the manuscript. The additional table further improves the quality of the paper.

I have no additional comments.

Reviewer #4 (Remarks to the Author):

I carefully read the revised manuscript and the authors' reply to all the issues raised.

I have no further comments.

The manuscript is worth to be published